# Selection of Olduvai Domains during Evolution: A Role for Primate-Specific Splicing Super-Enhancer and RNA Guanine Quadruplex in Bipartite *NBPF* Exons

**DOI:** 10.3390/brainsci12070874

**Published:** 2022-06-30

**Authors:** Igor Vořechovský

**Affiliations:** Faculty of Medicine, University of Southampton, HDH, MP808, Southampton SO16 6YD, UK; igvo@soton.ac.uk; Tel.: +44-2381-206425

**Keywords:** brain evolution, autism, neurons, brain size, NBPF, DUF1220, Olduvai, repeat, RNA guanine quadruplex, pre-mRNA splicing

## Abstract

Olduvai protein domains (also known as DUF1220 or NBPF) have undergone the greatest human-specific increase in the copy number of any coding region in the genome. Their repeat number was strongly associated with the evolutionary expansion of brain volumes, neuron counts and cognitive abilities, as well as with disorders of the autistic spectrum. Nevertheless, the domain function and cellular mechanisms underlying the positive selection of Olduvai DNA sequences in higher primates remain obscure. Here, I show that the inclusion of Olduvai exon doublets in mature transcripts is facilitated by a potent splicing enhancer that was created through duplication within the first exon. The enhancer is the strongest among the *NBPF* transcripts and further promotes the already high splicing activity of the unexpanded first exons of the two-exon domains, safeguarding the expanded Olduvai exon doublets in the mature transcriptome. The duplication also creates a predicted RNA guanine quadruplex that may regulate the access to spliceosomal components of the super-enhancer and influence the splicing of adjacent exons. Thus, positive Olduvai selection during primate evolution is likely to result from a combination of multiple targets in gene expression pathways, including RNA splicing.

## 1. Introduction

Gene sequences encoding Olduvai protein domains have undergone the largest human-specific increase in the copy number of any coding region in the genome [1,2]. In a recent genome assembly (hg38), their number exceeded 300 copies. Almost all of them have been inserted since the first emergence of Olduvai domains in non-primate mammals, with over half added to the human genome since the split of the *Homo* and *Pan* genera [2,3]. The Olduvai copy number increase was associated with the evolutionary expansion of the brain and grey matter volumes, neuron counts and cognitive abilities [1,3], as well as with disorders of the autistic spectrum and their severity [4,5]. Olduvai domains are highly expressed in neocortex neurons in regions believed to be critical to advanced cognitive functions [1]. Despite their prominent role in structural genome variation in health and disease and a robust positive selection favouring the Olduvai copy gains over 5–7 million years of human evolution [1,6], it remains a mystery why this repeat expansion has been safeguarded in the human genome and transcriptome and which cellular mechanisms served as targets for Olduvai selection in higher primates.

Olduvai domains are, on average, ~65 amino acids in length [1,7]. Based on their sequence similarity, the domains were categorised into six clades or sub-types: the conserved (CON1-3) sub-types can be present in non-primate mammals, whereas the HLS1-3 sub-types are human-lineage-specific [2]. Their order in NBPF proteins, where most Olduvai domains were found, was conserved: one or more CON1 clades at the N-terminus, followed by a single CON2 (except for *NBPF2P*), one or more HLS1-3 triplets and a C-terminal CON3 sub-type [2,6,7]. Each sub-type is encoded by an exon doublet (Figure 1). The first exons of the doublet are small, most often 52 nucleotides (nts), whereas the second exons are larger, exceeding the average size of human exons (~140 nts). The majority of *NBPF* introns remain small, consistent with their recent evolutionary history. However, the first exons in the expanded HLS1 sub-types are long (109 nts), as they accommodate a genomic insertion containing a guanine- and adenine-rich motif (denoted pG4), which was shown to form a DNA guanine quadruplex (G4) in vitro [6] (Figure 1B,C). In *NBPF* genes, the presence of pG4 perfectly discriminates between expanded (58/58) and unexpanded (0/12) HLS triplets [6]. In other words, when pG4 is present only in the penultimate exon of C-terminal CON3 sub-types, the expansion of pG4-lacking HLS triplets does not take place or is not selected. The HLS expansion by pG4 is selected only when pG4 is present in internal *NBPF* exons. Why? This question may be critical to our understanding of Olduvai evolution and function, and I propose a possible answer in a hypothesis formulated below.

## 2. Hypothesis

### 2.1. Giant Exonic Splicing Enhancers Created by an Intra-Exon Duplication of a Purine-Rich Motif in the Expanded HLS Exons

G4s arestable non-canonical secondary structures in DNA or RNA that are formed by the in-plane arrangement of four hydrogen-bonded guanines known as tetrads stacked one on top of another (reviewed in [12]). They have been suggested to influence the stability of Olduvai repeats and potentially drive structural DNA variation in the extant human population [6], but they could also act at the level of RNA processing and other gene expression steps [12]. To address the question if pG4 can act as an exonic splicing enhancer (ESE) and assess its capacity to promote exon inclusion in mature transcripts, each nucleotide in Olduvai exon doublets was assigned a hexamer ESEseq and ESSseq (ESS for exonic splicing silencer) score (Figure 1A). The scores, which are expressed here as ESE/ESSseq ratios, estimate the strength of exon selection and predict the exon–intron architecture by specifying a gradient of exon and intron definitions in primary transcripts [8,13]. In addition, the ESE/ESS and their scores carry a significant protein [14] and protein domain-related [9,15] information.

This analysis showed that the first exons of HLS doublets had a higher average ESE/ESSseq score than control human exons, both across unexpanded (pG4-) and expanded (pG4+) versions (Figure 1A). The maximum activity was observed in their middle parts, reaching the highest ESE/ESSseq score values in the expanded exons on three occasions, twice in the pG4 motif itself (Figure 1A). In contrast, the pG4-lacking versions of HLS1, as well as the HLS2 or HLS3 sub-types, showed only one or two ESE/ESSseq score peaks (Figure 1A). A closer inspection of their sequences showed that the extra peak was created by a duplication event and was co-localized with a short purine-rich motif that separated the duplicated regions (horizontal boxes in the top panel of Figure 1A). The mean ESE/ESSseq scores across pG4 were the highest along the *NBPF* transcripts and were significantly higher than their average values (0.25 vs. 0.05, *p* < 0.0001, unpaired *t*-test; Figure 1D, *top*). The ESE/ESSseq profiles in the first exon of the HLS1 doublets were, thus, distinct from the first exon of a CON sub-type, where most splice-enhancing activity was found in the vicinity of the 3′ splice site (*cf.* Figure S9 in [9]).

Taken together, these data showed that the splicing of expanded HLS1 exons was supported by a potent ESE that was created by intraexon duplication. The insertion of a pG4-containing purine-rich motif, thus, further enhanced the already high inclusion potential of the unexpanded first exons of the Olduvai exon doublets. The expected consequence of the CON3 > HLS1 evolutionary scenario [6] (Figure 1C) would be an improved exon selection, facilitating the retention of expanded HLS1 triplets in the mature transcriptome and serving as a selection target for Olduvai expansions at the RNA level.

### 2.2. Intragenic Duplications of Olduvai Exon Doublets and Their Possible Impact on Alternative Splicing

Small exons (~60 nts or less) have been associated with poor inclusion in messenger RNAs [16,17,18,19]. This length limit also applies to constitutively spliced internal exons, which have an optimal size range between 60 and 200 nts [19]. Although smaller exons may lack cross-exon interactions with spliceosomal components, many mini- or even micro-exons are efficiently recognised in vivo in a constitutively or alternatively spliced manner [17,20]. Such exons are often located very close to additional exons upstream or downstream that are separated by short introns withregulatory functions [17]. The requirement for flanking exons disappeared when mini-exons were expanded [17]. The splicing of an intron was also enhanced when coupled with the splicing of a downstream intron, possibly through a mechanism independent of exon junction complex depositions [19], further adding to the evidence that individual exons are not selected independently. These observations are in line with Nature’s experiments, showing that splice-site mutations in human disease genes may not affect the splicing of only the mutated exon, but also adjacent exons or introns, most often leading to the skipping of one or more downstream exon [21,22,23,24,25,26]. Such splicing dependencies may be created by exon duplications into a new genomic context, such as those in the *NBPF* genes. The Olduvai domain duplications involved ~4.7 kbs regions, consisting of six-exon and six-intron blocks [2]. Based on these studies, one can hypothesize that the improved inclusion of a single internal HLS exon in mRNA through pG4 gain (Figure 1A) may alter the recognition of adjacent exons and introns and their native splicing patterns, potentially also contributing to their evolutionary spread. By contrast, penultimate pG4-containing *NBPF* exons of terminal CON3 sub-types would lack this property, preventing the selection of CON3 expansions in the absence of internal pG4-containing exons (Figure 1C).

Given the limitations of the next-generation RNA sequencing (RNA-seq) and other methods used to unambiguously identify *NBPF* exons in the human transcriptome (HLS exons are 96–100% identical, ref. [2]), our understanding of their alternative splicing patterns remains very limited [27]. Although the functions of distinct mRNA isoforms and Olduvai sub-types are unknown, systematic approaches employing an iterative deviation method for RNA-seq datasets from 16 human tissues identified 17 *NBPF* exons among 3100 ‘switch-like’ events [28]. Switch-like exons show a high usage in one tissue and low usage in another, suggesting that they are regulated [28]. However, the functional significance and exact identities of such alternatively spliced *NBPF* exons remain obscure, awaiting experimental confirmation.

### 2.3. Is Access to the Super-Enhancer in the First HLS1 Exons Regulated by RNA G4 Formation?

The 5′ part of the first exon encoding CON1 in *NBPF* genes resembles a trinucleotide repeat (Figure S9 in [9]). Trinucleotide-containing RNAs and other microsatellites can form assemblies of intracellular RNA aggregates that cause more than 30 genetic disorders by sequestrating RNA-binding proteins, leading to downstream changes in alternative splicing and spliceopathies [29]. For example, a GC-rich microsatellite expansion in the first *CNBP* intron in type 2 myotonic dystrophy can trigger intron retention [30]. Both the trinucleotide-like exonic segment [9] and the more recent pG4-led expansions of HLS1 sub-types [6] showed strong purine enrichment (Figure 1A). A nearly exclusive purine composition of pG4 is likely to diminish intramolecular Watson–Crick base-pairing in these regions and increase their RNA single-strandedness in the absence of non-canonical structures. Although the pG4-containing insertion (Figure 1B) does not meet most stringent requirements for G4 formation (G_≥3_N_1__–__7_G_≥3_N_1__–__7_G_≥3_N_1__–__7_G_≥3_, where G is guanine and N is any nucleotide), RNA G4 for this insertion was strongly predicted by multiple algorithms (Figure 1D), including those employing artificial neural networks without reliance on motif definitions [11]. This is in agreement with evidence for DNA G4 formation obtained by circular dichroism spectroscopy of pG4 [6]. Additionally, RNA G4 formation would be supported by a lack of cytosines in sequences flanking pG4, except for the 3′ terminus of the exon (Figure 1A). A single cytosine in the middle of pG4 (Figure 1B) is unlikely to interfere with RNA G4 folding. Thus, the pG4-led splice-enhancing activity of expanded HLS1 sub-types shown here may depend on RNA G4 and conformational switches between canonical and non-canonical RNA structures.

The link between RNA G4 formation and the regulation of RNA processing by ESEs is increasingly supported in the literature. For example, a high-purine G4 has been shown to enhance exon inclusion in *FMR1* [31]. In addition, G4-prone motifs in primary transcripts can activate heterologous exons if placed either downstream or upstream of the tested exon [32], although they may also inhibit splicing [12,33,34]. Interestingly, the strongest ESE hexamer AGAAGA was among those reported to require a partner motif or motifs within 16 nts flanking regions [8]. Such motifs may be present in the 33 nts duplication in the first HLS1 exon (Figure 1A),however, it remains to be seen to what extent such requirements reflect specific canonical or non-canonical RNA structures.

Finally, an underappreciated aspect of G4 biology is the intercalation of monovalent and divalent metal ions into the centre of or between G4 tetrads, stabilizing them in a metal-specific manner and enhancing their base-stacking interactions [35,36,37]. The evolution of the ESE and ESS was tailored by binding sites of divalent metals, such as Ca^2+^, reflecting their position in the Irving–Williams stability series [9,15]. Both mono- and divalent metal ions are pivotal for G4 stability gradients, which have been linked to their ionic radius, hydration energy and binding strength toward the guanine O^6^ [37]. For example, larger cations can coordinate eight oxygen atoms, while smaller ions coordinate only four, contributing to distinct G4 stabilities ([12,37] and references therein). Metal ions may also control structural switches between G4 and canonical conformations involving alternative RNA secondary structures in weakly paired regions, such as pG4. Ca^2+^ was shown to induce the structural transition of anti-parallel to parallel G4 via multiple steps [38] and also exert a strong stabilizing effect on guanine triplexes [39]. It would not be surprising if the metallome-dependent RNA G4 formation constituted a discernible selection force to shape the evolution of the auxiliary splicing code.

Taken together, the expanded and unexpanded first Olduvai exons provide a new and attractive model to test the interplay between ESE and RNA G4 and their respective trans-acting factors.

### 2.4. Conclusions

In summary, it was proposed that the rapid evolutionary spread of Olduvai domains in higher primates was facilitated by prominent splicing activities of expanded internal *NBPF* exons. Their positive selection at the exon-level is likely to depend on a combinatorial control by a strong splicing enhancer created by intraexon duplication and by the formation of the RNA G4 and/or stable intermolecular RNA:DNA hybrids during transcription. Thus, the selection of these enigmatic domains during recent evolution may have acted on multiple targets, including RNA processing (Figure 1), DNA stability [6] and at the protein level [40]. In future studies, it is important to establish the role of intraexon duplication in *NBPF* splicing experimentally, evaluate the impact of the proposed *NBPF* non-allelic homologous recombination event [6] (Figure 1C) on RNA processing and structure and fully characterize alternative RNA splicing of the *NBPF* genes in brain regions where it is also most prevalent [41]. Future studies on this interesting exon-expansion model may provide valuable insights into our understanding of human brain development and the acquisition of advanced cognitive functions.

## 3. Material and Methods

ESE/ESSseq scores were derived from exon inclusion levels measured by the RNA-seq of unspliced and spliced central minigene exons that contained comprehensive libraries of 4096 hexamers cloned at five different positions [8]. Briefly, the variant minigene library (input) was transfected into human embryonic kidney cells and 24 h later the messenger RNA molecules that had included the central exon (output) were isolated by size selection and sequenced [8]. For each hexamer, an enrichment of output proportion over input proportion was calculated and expressed as ESEseq and ESSseq scores [8]. Greater ESE/ESSseq score values indicated that the central variant exon was spliced more efficiently, i.e., they contained a larger excess of ESEs over ESSs [8,9,15]. The ESE/ESSseq values, thus, provided reasonable estimates of exon inclusion in mature transcripts and their splicing activities [8]. The scores were then assigned to each nucleotide position of the *NBPF* transcripts, as described in detail for exonic sequences that encode Ca^2+^-binding sites [9]. The list of hexamers with combinatorial requirements for flanking RNA motifs or structures was reported by Ke et al. [8].

The prediction of RNA G4 structures was carried out using default options of the G4RNA screener [11] and the indicated pG4-containing sequence of *NBPF9* as an input (Figure 1D). Threshold values are specified in Figure 1 legend. Alignments (Figure 1) were created with Clustal Omega (v. 1.2.4) [42] (www.ebi.ac.uk/Tools/msa/clustalo/, accessed on 3 May 2022) using Ensembl [43] (build 104; www.ensembl.org, accessed on 3 May 2022) *NBPF* gene sequences.

## Figures and Tables

**Figure 1 brainsci-12-00874-f001:**
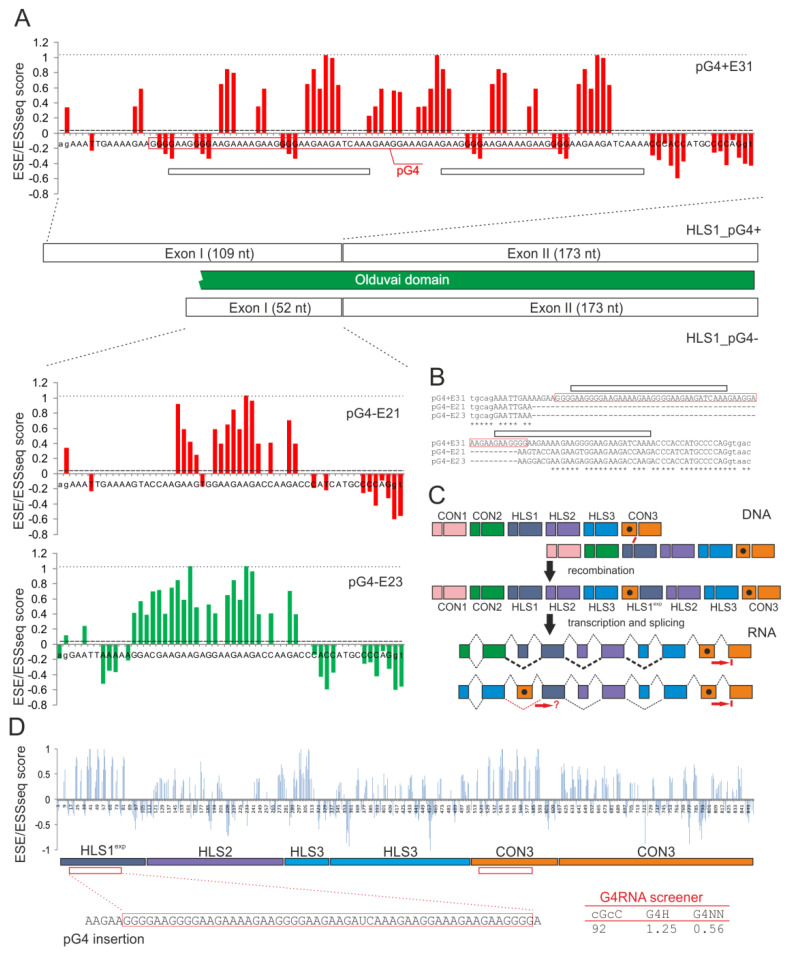
Splicing enhancer activities of the first exon in expanded and unexpanded Olduvai exon doublets and RNA G4 predictions. (**A**) ESE/ESSseq scores for overlapping hexamers in representative pG4-containing (pG4+, *top panel*) and pG4-lacking (pG4-, *bottom panel*) *NBPF10* exons. Exon (E) numbers correspond to the longest transcripts, with E21 and E23 representing examples of HLS sub-types with unexpanded first exons of the Olduvai doublet and E31 representing the expanded version. Horizontal dotted lines at the top of each chart denote the maximum ESE/ESSseq values (1.034 for the strongest splicing hexamer AGAAGA, ref. [8]). The horizontal dashed line shows mean values for control human exons [9]. Horizontal black boxes at the top panel denote duplicated regions in the expanded first exons of Olduvai doublets. The 68 nt pG4 sequence [6] is in a red box. (**B**) The alignment of the *NBPF10* exons that contain (+) or lack (−) pG4. Exonic sequences are in upper case and flanking intronic sequences are in lower case. pG4 is in a red box; black boxes denote duplicated regions. (**C**) Predicted effects of Olduvai exon expansions on RNA processing. *Top*, non-homologous allelic recombination was previously proposed to explain Olduvai amplifications [6]. Black dots denote pG4 sequences, red line denotes the location of an intron recombination breakpoint [6]. ^exp^, expanded Olduvai sub-type. *Bottom*, the putative impact of the recombination event on pre-mRNA splicing. Splicing is shown as diagonal lines for canonical (dotted lines) or alternative (dashed lines) events; the line widths correspond to expected exon usage frequencies. Red arrows at penultimate exons illustrate a lack of splicing dependencies downstream of pG4-containing exons in CON3. Retention of the last intron might also lead to stable translation of truncated proteins, bypassing nonsense-mediated RNA decay of transcripts with premature termination codons further upstream [10]. (**D**) The ESE/ESSseq profile in an *NBPF9* region between the first expanded HLS1 exon and the terminal CON3 exon (*top*). Nt numbering is from the first position of the expanded HLS1 exon. *Bottom*, pG4-containing insertion in HLS1 predicted to form RNA G4 by the indicated methods. Score predictions were carried out with the G4RNA screener [11] (v. 0.2, window length 60, window step 10). Thresholds for the consecutive guanine over consecutive cytosine (cGcC) scores were >4.5, for the Genehunter (G4H) scores were >0.9 and for the neural network (G4NN) scores were >0.5. Each method identified the RNA G4 structures in the absence of ligands.

## Data Availability

Not applicable.

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
