# Peer review of "Selection of Olduvai Domains during Evolution: A Role for Primate-Specific Splicing Super-Enhancer and RNA Guanine Quadruplex in Bipartite *NBPF* Exons"

_brainsci, 2022, doi:10.3390/brainsci12070874_

Round 1
Reviewer 1 Report
My suggestions:
1. In the introduction I would add a schematic figure on the differences between the three CON and HLS subtypes.
2. Can the intragenic duplications in Olduvai exon impact some kind of phenotypical diversity in humans?
3. Does the abnormally spliced or under/overexpressed Olduvai exon impact some kind of disease?
4. Is there 3D model available on protein structure Olduvai domains? If yes, it would clearly improve the manuscript.
5. Were there any interacting partners of Olduvai domains described? A brief explanation may be useful.
Reviewer 2 Report
The manuscript entitled, ‘Selection of Olduvai domains during evolution: a role for primate-specific splicing super enhancer and RNA guanine quadruplex in bipartite NBPF exons’ is a hypothesis paper focused on olduvai protein domain, DUF1220, and NBPF, suggesting the exon doublets in mature transcripts are facilitated by enhanced duplicated splice variants within the first exon. This creates a predicted RNA guanine quadruplex that influences splicing of adjacent exons enabling expansion of olduvai exon doublets having a role in neural development, which may reflect on human brain functions.
Specific comments:
The hypothesis is comprehensive. How would the author predict the functional role of these repeat splice variants? Although few studies have suggested its role in neurodevelopment, do these tandem repeats of splice variants influence cognition?
Author Response
See attached file for point-by-point responses to each reviewer

Reviewer 3 Report
The manuscript entitled “Selection of Olduvai domains during evolution: a role for primate-specific splicing superenhancer and RNA guanine quadruplex in bipartite NBPF exons” By Igor Vorechovsky is scientifically interesting. However it needs revision.
Title is not reflecting the concept of “hypothesis”, it would be important to differentiate the reseach papers from “a hypothesis” , hence the title shall be revived with the term “a hypothesis” like “Selection of Olduvai domains during evolution: a role for primate-specific splicing superenhancer and RNA guanine quadruplex in bipartite NBPF exons: a hypothesis”
Major concern on the figure 1 is author has not cite any article for the source of the images. However, the details “Although the pG4-containing insertion (Fig. 1B) does not meet stringent requirements for G4 formation (G=3N1-7G=3N1-7G=3N1-7G=3, where G is guanine and N any nucleo- tide), RNA G4 for this insertion is strongly predicted by multiple algorithms (Fig. 1D), including those employing artificial neural network without reliance on motif definition [28], in line with the experimental proof for G4 formation by the pG4 DNA [5]. The pG4 flanking sequences lack cyto- sines, except for the 3’ terminus of the exon. This would promote RNA G4 folding [28 and refer- ences therein] while a single cytosine in the middle of pG4 (Fig. 1B)” looks like it is sourced from various reference. In case it is an original drawing of the author, it would be ok, if not it would be important to cite the sources in the legend “Splicing enhancer activities of the first exon in expanded and unexpanded Olduvai exon doublets and RNA G4 predictions”
Author should directly reach the conclusion of the hypothesis. Details of previous findings shall be added in the earlier sections.
Defined statement of hypothesis is missing in the entire document. It would be appropriate to state the exact hypothesis in the conclusion.
Material and Methods is too short with no details. Author shall add details to support and for the benefit of readers. Statement like “Alignments were created with Clustal Omega (1.2.4) using Ensembl (build 104) gene sequences.” is not self-explanatory and too short.
Round 2
Reviewer 1 Report
The authors fulfilled my suggestions.
Reviewer 3 Report
Revised MS can be accepted